# Plant Response to Mechanically-Induced Stress: A Case Study on Specialized Metabolites of Leafy Vegetables

**DOI:** 10.3390/plants10122650

**Published:** 2021-12-02

**Authors:** Jana Šic Žlabur, Sanja Radman, Sanja Fabek Uher, Nevena Opačić, Božidar Benko, Ante Galić, Paola Samirić, Sandra Voća

**Affiliations:** 1Department of Agricultural Technology, Storage and Transport, University of Zagreb Faculty of Agriculture, Svetošimunska Cesta 25, 10000 Zagreb, Croatia; jszlabur@agr.hr (J.Š.Ž.); agalic@agr.hr (A.G.); paolasamiric@hotmail.com (P.S.); svoca@agr.hr (S.V.); 2Department of Vegetable Crops, University of Zagreb Faculty of Agriculture, Svetošimunska Cesta 25, 10000 Zagreb, Croatia; sfabek@agr.hr (S.F.U.); nopacic@agr.hr (N.O.); bbenko@agr.hr (B.B.)

**Keywords:** brushing, lettuce, chicory, phytochemicals, antioxidant capacity

## Abstract

Plants have evolved various adaptive mechanisms to environmental stresses, such as sensory mechanisms to detect mechanical stimuli. This plant adaptation has been successfully used in the production practice of leafy vegetables, called mechanical conditioning, for many years, but there is still a lack of research on the effects of mechanically-induced stress on the content of specialized metabolites, or phytochemicals with significant antioxidant activity. Therefore, the aim of this study was to determine the content of specialized metabolites and antioxidant capacity of lettuce and green chicory under the influence of mechanical stimulation by brushing. Mechanically-induced stress had a positive effect on the content of major antioxidants in plant cells, specifically vitamin C, total phenols, and flavonoids. In contrast, no effect of mechanical stimulation was found on the content of pigments, total chlorophylls, and carotenoids. Based on the obtained results, it can be concluded that induced mechanical stress is a good practice in the cultivation of leafy vegetables, the application of which provides high quality plant material with high nutritional potential and significantly higher content of antioxidants and phytochemicals important for human health.

## 1. Introduction

Consumer interest in cut leafy vegetables with distinct functional value is increasing, as is the need for a continuous supply to the market. Growing leafy vegetables in greenhouses allows them to be grown throughout the year, even in the cold months, thus ensuring the supply and availability of various leafy vegetables, especially lettuce, in the off-season [1,2]. Recently, the cultivation of leafy vegetables in greenhouses has been increasingly oriented towards hydroponic cultivation, mainly because of a number of advantages that this type of cultivation has over the conventional one: significant yield, high quality and healthy plant material, lower incidence of pathogens, less use of pesticides, less pollution, conservation of groundwater (closed hydroponic systems), high degree of automation, less physical labor, better control of water and nutrient supply to plants, fewer weeds, etc. [3,4]. Floating hydroponics (FH) is suitable for growing leafy vegetables such as lettuce, spinach, chicory, arugula, lettuce, medicinal, and also aromatic plants. The benefits are many and include faster growth, earlier harvest, more production cycles, and higher yield per unit area due to better control of plant nutrients [5,6]. Due to the higher plant density in FH and the lack of sunlight or the change in its spectrum during the winter season, plants compete for light, resulting in an undesirable elongation of the hypocotyl and internodes of the stem, leading to fragile plants, poor quality and uneven growth [2,7]. To produce the strongest and most resistant plant, different treatments can be combined to strengthen it. Mechanical conditioning (MC), i.e., stimulation of plants by tactile stimuli in the form of touching, brushing, or rubbing the plant material, is a successful, non-invasive, environmentally friendly, simple, and inexpensive measure to regulate plant growth that can reduce elongation and increase plant strength and resistance [8,9]. Indeed, during growth, plants are exposed to various abiotic and biotic stresses such as wind, rain, machinery, animals, pathogens, or plants themselves [10,11,12]. Plants have evolved numerous adaptations to these stresses and trigger a range of responses including anatomical, physiological, biochemical, and molecular, known as thigmomorphogenesis [13,14]. This plant adaptation is used to develop a successful production practice known as mechanical conditioning. This is based on the fact that mechanically-induced stress (MIS), applied naturally or under controlled conditions, impairs growth, thereby reducing the mass and size of major plant parts [15]. One of the most noticeable effects of MIS on plants is a reduction in the length of stems, leaves or petioles, resulting in plants that are smaller and more compact than those grown in unstressed controls, i.e., a reduction in primary shoot growth and an increase in secondary thickening [10,11,16,17,18].

However, it is important to emphasize that MIS is not necessarily associated with injury (wounding), but plants respond by activating its defense mechanism. Namely, the changes in the wax layer lead to the induction of so-called touch-sensitive genes and membrane receptors, whereupon the plant activates its defense mechanisms [19]. However, the response of plants to MIS in some variables other than morphological, such as the content of specific phytochemicals (chlorophyll, vitamins, phenols, etc.) is highly dependent on the species, and there are a number of unanswered questions [10,18].

Due to its positive effects on morphological properties [10,11,12,15,20], the MC technique has been successfully used for many years in winter production of various leafy vegetables and medicinal and aromatic plants, but there is still a lack of research on the effects of MIS on the chemical composition and content of specialized metabolites with significant antioxidant activity in plant material. Therefore, the aim of this study was to determine the content of specialized metabolites and antioxidant capacity of lettuce and green chicory under the influence of mechanical stimulation by brushing in a hydroponic growing system.

## 2. Results

### 2.1. Specialized Metabolites Content of Lettuce and Chicory

The results of the analyzed specialized metabolites of lettuce are presented in Table 1 and Table 2. According to the results, the significant effect of MIS on ascorbic acid (AsA) content in lettuce (Table 1) is most pronounced in the first harvest period (L-MC10_1_, L-MC20_1_), with the highest AsA content (24.39 mg/100 g fw) determined in a sample treated with 20 brushings per day (L-MC20_1_). In the first harvest period, an average of 57% higher AsA content was determined in treatment L-MC20_1_ than in the treatment in which the procedure MC was not applied (L-MC0_1_). In the same harvest period, the lettuce plants treated with MC with 20 passes per day also had the highest total phenol (TPC) content determined (102.46 mg GAE/100 g fw); even several times more than the control sample (L-MC0_1_) and 14% more than the treatment with 10 passes per day (L-MC10_1_). Total flavonoid (TFC) content in lettuce plants treated with 20 passes per day (L-MC20_2_) was also highest; 87% higher than the control sample and 21% higher than the 10 passes per day treatment. In the second lettuce harvest period, the effect of mechanical stimulation on AsA content was not statistically justified, i.e., no significant differences were observed between treatments, although a slight increase in AsA content was observed in the treatment with 10 passes per day (L-MC10_2_), about 7% compared to the sample without treatment (L-MC0_2_). Significantly higher values of TPC, TFC and total non-flavonoid (TNFC) were observed in the second harvest period compared to the content of polyphenolic compounds in the first harvest period, with the highest TPC in the sample treated with 10 passes (L-MC10_2_) and the highest TFC in the treatment with 20 passes (L-MC20_2_). The highest levels of total chlorophyll (TCh) and carotenoids (TCA) in lettuce (Table 2) in the first harvest period were found in the samples treated with MC by 20 brushes per day, but in the second harvest the highest TCh and TCA were noted in plants MC treated by 10 brushes per day.

Green chicory samples (Table 3) were again found to have higher AsA content in the first harvest, regardless of the treatment applied. The influence of MC showed a slightly different trend in AsA content than in the lettuce samples. Indeed, the highest AsA content (40.97 mg/100 g fw) was detected in the 10 brush strokes per day treatment (GC-MC10_1_), with a 12% higher AsA content compared to the control sample (GC-MC0_1_) and 19% higher compared to the 20 brush strokes per day treatment (GC-MC20_1_). A significantly lower AsA content was observed in the second harvest period than in the first. However, in the second harvest period, the highest AsA content (33.23 mg/100 g weight) was detected in a sample treated with 20 brush strokes per day (GC-MC20_2_), by an average of 32% compared to the treatment with 10 brush strokes (GC-MC10_2_) and 41% compared to the control sample (GC-MC0_1_). The highest levels of TPC (179.77 mg GAE/100 g fw), TFC (103.14 mg CTH/100 g fw) and TNFC (76.63 mg GAE/100 g fw) were found in the 20-pass treatment (GC-MC20_2_) in the second harvest period. In general, as in lettuce samples, regardless of the treatment applied, the highest levels of polyphenolic compounds were determined in chicory in the second harvest period. Regarding the analyzed total chlorophylls content in green chicory (Table 4), no significant differences were found depending on the MC treatment (10 and 20 brushes per day) applied both in the first and second harvest period, while TCA content differed significantly considering the brushing treatment with the highest determined content in the first harvest period (0.30 mg/g) in plants treated with 10 passes per day (GC-MC10_1_) and 0.48 mg/g in second harvest period. In general, the highest TCh and TCA contents were determined in the second harvest period with average values of 0.89 mg/g for TCh and 0.43 mg/g for TCA, which means about 62% higher TCh contents and about 65% higher TCA contents compared to the average values of TCh and TCA found in the first harvest period.

### 2.2. Antioxidant Capacity of Lettuce and Chicory

The results of antioxidant capacity of lettuce and green chicory samples are shown in Figure 1 and Figure 2. In general, relatively high values of antioxidant capacity were obtained in the first harvesting period, regardless of the treatment. The highest antioxidant capacity (2267.55 µmol TE/L) was determined in a lettuce sample treated with 20 brush passes per day (L-MC20_1_), i.e., in plants with the most pronounced MIS. The same trend of antioxidant capacity of lettuce was observed in the second harvest period considering MC treatment; the highest antioxidant capacity was determined in the sample with 20 passes per day (L-MC20_2_) (Figure 1). In the green chicory samples (Figure 2), the variations in antioxidant capacity are small compared to the lettuce samples, since no significant statistical differences were observed in either the first or second harvest period when treated with MC. However, regardless of MIS treatment and harvest period, green chicory samples are characterized by high antioxidant capacity, indicating a nutritionally valuable plant material.

## 3. Discussion

### 3.1. Specialized Metabolites Content

During growth, plants are exposed to many different stress factors (biotic and abiotic) and have developed numerous protective mechanisms, i.e., responses to them. The direct response of plants to environmental stress is increased production of reactive oxygen species (ROS). This is because ROS play an important role as a signaling molecule in regulating plant growth and enhancing plant responses to stress. In general, ROS are considered as by-products of aerobic metabolism in plants and are produced in various cellular organelles such as chloroplasts, mitochondria and peroxisomes. Plants require a threshold level of ROS for their vital functions, and change in concentration alters the physiology of plants, disrupting cell metabolism and causing irreversible damage to DNA material. Precisely to maintain the balance of ROS in cells, plants have also developed an antioxidant defense system (enzymatic and non-enzymatic mechanisms) that maintains a redox state of cells that helps to eliminate ROS [21,22,23].

Despite the lack of scientific research on the effect and correlation of MIS on AsA content, the effect of mechanical stimuli on AsA content can be explained by metabolic reactions that occur in plant tissues (cells) when the plant is exposed to stress. The non-enzymatic mechanisms of plant defense against ROS are mainly mediated by low molecular weight antioxidants such as glutathione, AsA and flavonoids, which are known to remove hydroxyl radicals and singlet oxygen. Ascorbic acid is indeed one of the best-known oxygen scavenging molecules, i.e., AsA protects metabolic processes from the harmful effects of H_2_O_2_ and other toxic oxygen-derived radicals. An important segment of enzymatic protection is the response of plants via enzymatic mechanisms (e.g., the enzyme ascorbate peroxidase, APX), such as another important ROS scavenging pathway, namely the Foyer–Halliwell–Asada pathway, also known as the ascorbate–glutathione cycle, which occurs in chloroplasts, mitochondria, apoplasts, cytosol, and peroxisomes. The glutathione–ascorbate cycle is a metabolic pathway for the de-toxification of hydrogen peroxide (H_2_O_2_), a reactive oxygen species produced as a byproduct in plant metabolism. The cycle involves the antioxidant metabolites ascorbate, glutathione, and NADPH, and the enzymes that link these metabolites [22,23]. Ascorbate peroxidase (APX), glutathione reductase (GR), monodehydroascorbate reductase (MDHAR) and dehydroascorbate reductase (DHAR) are key enzymes related to the non-enzymatic antioxidant metabolites AsA and glutathione. Dehydroxyascorbate (DHA), which is itself an oxidized form of ascorbate, is also formed from the eponymous cycle [24]. The ascorbate–glutathione cycle plays an important role in the reduction of dehydroxyascorbate to ascorbic acid [24]. Considering all this, it can be confirmed that AsA protects metabolic processes from the harmful effects of H_2_O_2_ and other toxic radicals from oxygen as well as membranes, either by directly removing toxic radicals such as ^1^O_2_, O_2_- and OH- or indirectly through the regeneration of the reduced forms of tocopherol or zeaxanthin. In addition, numerous studies show that plants exposed to various stress conditions, especially drought [25,26] and salinity [27,28], had increased AsA content. If we relate all this to the results of this study, it is worth noting that higher AsA content was found in lettuce in the first harvest period when plants were exposed to MIS, regardless of the brush strokes per day, while no significant changes were found in the second harvest period given the MIS treatment. In green chicory, in the first harvest period, the highest AsA content was found in plants treated with 10 brush strokes per day, while in the second harvest period, plants were treated with 20 brush strokes per day. (Table 1 and Table 3). Comparing the AsA content in non-mechanically treated lettuce (control sample) with the results of the study by the authors Medina-Lozano et al. [29], we can generally conclude that slightly higher values were obtained in this study; however, compared to the study by van Treuren et al. [30], lower values were found in this study. Furthermore, the AsA content in the control green chicory samples ((GC-MC0), without mechanical treatment) analyzed in this study was higher than that reported in other literature data [31,32]. The reason for this wide dispersion of results is mainly due to the genetic characteristics of the varieties themselves, which are most commonly used for cultivation for commercial purposes [29,30,31,32].

The MIS, caused by the number of brushings, induces mechanical depolarization of the plant cell membrane, leading to the formation of various types of free radicals and initiating lipid peroxidation, thereby activating plant protective mechanisms against ROS, i.e., increased activity of antioxidant mechanisms and precursor enzymes of vitamin C synthesis (i.e., APX) [13,33]. Nowadays, among the best known and studied antioxidants present in cellular organelles and whose main function is detoxification of ROS are the polyphenolic compounds. Apart from being among the compounds with the most potent antioxidant activity, polyphenolic compounds form a chemically extremely large, diverse and highly significant group of plant secondary metabolites. Phenolic compounds are in fact the main products of the plant defense system united in secondary metabolism, i.e., the processes of biosynthesis of phenylpropanoids, anthocyanins, alkaloids, coumarins, terpenes, tannins, glucosinolates, flavonoids and isoflavonoids, lignans and lignans, among others. Secondary metabolites have several important ecological and physiological functions in the plant organism, ranging from structural roles (binding of cell wall polysaccharides), protection from attack by herbivores and microorganisms, attraction of pollinators, seed dispersal, communication of the plant with other plants and organisms, to the most important role, the responses of the plant organism to stress. Indeed, the phenylpropanoid biosynthetic pathway, leading to the accumulation of various phenolics compounds, is activated in plants under stressful environmental conditions. Thus, the accumulation of polyphenolic compounds can be regulated by various environmental factors such as water availability, wounding, herbicides, insect herbivory, nutrient deficiency and availability, light exposure, salinity, developmental stage of the plant, etc. [14,22,34,35]. Numerous scientific studies indicate increased accumulation of polyphenolic compounds when a plant organism is exposed to stress, as a direct response by the plant to defend itself. Mechanical conditioning also exposes the plant to mechanical stress, resulting in changes in metabolic pathways and increased activation of compounds whose main function is to remove excessive amounts of ROS, accumulated in plant cells [14]. Indeed, plants respond to MIS with stress responses such as the aforementioned increased ROS production and changes in calcium levels, followed by the activation of defense responses [12]. As mentioned above, polyphenols are one of the most effective compounds in the elimination of free radicals, and it is therefore expected that when the plant is exposed to stress, an increased accumulation of these compounds begins [36], which is confirmed by the results of this research. Lettuce and chicory plants that were more mechanically stimulated (Table 1 and Table 2), i.e., exposed to MIS, also showed higher levels of polyphenolic compounds, especially total phenols and flavonoids. It should be emphasized that regardless of the mechanical treatment, i.e., MIS, the samples of lettuce and green chicory are rich in polyphenolic compounds and the results of total phenolics (TPC) are generally in agreement with other literature data, with the values of polyphenolic compounds obtained in this study for lettuce and green chicory not mechanically treated (control samples) similar or slightly lower than the data reported by other authors [36,37,38,39,40]. Based on the results obtained, it can be concluded that mechanical stimulation induces plant responses in direct response to stress exposure, as evidenced by the increased content of polyphenolic compounds in the lettuce and chicory samples in the MC treatments. The results of the higher content of polyphenolic compounds in plants exposed to stress are in agreement with the statements of other authors who also confirm that plants have the ability to biosynthesize higher amounts of polyphenols for defense purposes under stress than under normal growing conditions. Indeed, the biosynthesis of polyphenols in plants under stress conditions is regulated by different activities of specific enzymes in the metabolic pathways of polyphenol synthesis (shikimate/phenylpropanoid and mevalonate pathways) such as phenylalanine ammonia-lyase (PAL) and chalcone synthase (CHS). Under stress conditions, there are enhanced effects of enzymes associated with the regulation of gene transcription, which encode important biosynthetic enzymes of polyphenolic compounds [14,36].

Stress conditions to which plants are frequently exposed during their growth, especially those caused by abiotic factors such as heavy metals, drought, too much light (UV radiation), too low or too high temperatures, increased soil salinity, etc., usually have a negative effect on the performance of the photosynthetic process. These stress conditions reduce stomatal conduction, which leads to oxidative stress and reduces the activity of RUBISCO enzymes, which prevents the smooth process of photosynthesis by negatively affecting photosystem I and II, photosynthetic transport electron and finally chlorophyll biosynthesis [41]. The authors Benikhlef et al. [12] in their study on the induction of soft mechanical stress in plants of the species *Arabidopsis thalian* come to different conclusions. In fact, the aforementioned authors note that the application of MIS did not result in obvious damage to plant tissues, while chlorophyll content increased, which is directly related to the rapid change in calcium concentration and the release of ROS, accompanied by changes in cuticle permeability, induction of gene expression typically associated with mechanical stress, and the release of biologically active diffusers from the surface. The results of the pigment compounds analyzed in this study are mostly in agreement with the results of the mentioned study. In the lettuce samples (Table 3), both TCh and TCA content were highest in the first harvest period when MC was applied with 20 passes per day and 10 passes per day in the second harvest period. However, this effect of MIS on pigment compounds was not as pronounced in the green chicory samples, as there were no significant differences between the controls and those treated with MC. Considering that both species have a characteristic green color, high levels of total chlorophylls and carotenoids were also expected in the control samples, i.e., those not subjected to the MIS. The results of total chlorophylls and carotenoids content of the green chicory and lettuce samples from this study are significantly higher than those found in other research [32,42,43].

### 3.2. Antioxidant Capacity

Accumulation of antioxidants in plant cells is the expected response of the plant defense mechanism, wherein the bioactive compounds are those that are intensified to synthesize in order to detoxify ROS. From bioactive compounds, one that exhibits the most significant antioxidant activity in the plant cell are polyphenolic compounds, such as flavonoids [9,14,21,22,23,34,35,36]. It is also important to emphasize that antioxidants, in addition to the important effect and protection of plant cells, also have a beneficial effect on human health, given that their mechanisms of action, such as radical scavengers, chelators, quenchers, oxygen scavengers and antioxidant regenerators, effectively inhibit free radicals accumulated in human cells as a result of oxidative processes [44,45,46]. Based on the above mentioned and obtained results, it can be concluded that that cultivated leafy vegetables abound antioxidants and thus present a high value nutritional raw material. According to other studies, the lettuce and green chicory species analyzed in this study, regardless of MIS, are vegetables that have significant antioxidant properties, i.e., high levels of antioxidant capacity [32,37,47].

## 4. Materials and Methods

### 4.1. Plant Material

The research was conducted in 2019, at the Experimental Station of the Department of Vegetable Crops at the Faculty of Agriculture, University of Zagreb. Two types of green leafy vegetables were grown for research purposes: lettuce (*Lactuca sativa* L.) and green chicory (*Cichorium intibus* var. *foliosum*). Of the *Lactuca sativa* species, a mixed lettuce was sown in the following composition: ‘Reggina di Maggio’ 25%; ‘Meraviglia delle Quatro Stagioni’ 25%; ‘Grunetta’ 25%; ‘Cavolo di Napoli’ 25% (Hortus sementi, Longiano, Italy), while of the green chicory, the variety ‘Zuccherina di Trieste’ (Hortus sementi, Longiano, Italy) was sown.

### 4.2. Floating Hydroponics

Both leafy vegetables species were grown in floating hydroponics in an unheated greenhouse (45°49′33.208″ N, 16°1′42.832″ E). A total of 12 polystyrene boards filled with inert perlite substrate (Europerl d.o.o., Samobor, Croatia) were used for growing lettuce and chicory in the hydroponic system, of which six plates (two for each of the treatments, Figure 1) were sown with lettuce seeds and six with chicory seeds. Manual sowing of lettuce and chicory in an unheated greenhouse was carried out on 23 April. For both species, 30 seeds were sown per slot of the 0.96 m × 0.60 m board with a total of 102 slots (17 cm long and 0.5 cm wide). The amount of seed required depended on the seed size of the species, 22.77 g/m^2^ for chicory and 10.35 g/m^2^ for lettuce. After sowing, the seeds were covered with a finer granulation of perlite (0–3 mm) and the moisture of the substrate was maintained during germination. After emergence, the plates were placed in basins (4 m × 2 m × 0.25 m) filled with nutrient solution. The nutrient solution used in the experiment was adapted for growing lettuce and chicory [48]. The contents of the nutrient solution used for hydroponic cultivation of lettuce and chicory are shown in Table 5.

### 4.3. Mechanical Conditioning

The mechanical conditioning process for both vegetable species began in early May, from the appearance of the cotyledon till harvests. The process was carried out by brushing with a burlap cloth in two mechanical conditioning treatments: 10 (MC10 treatment) and 20 (MC20 treatment) passes per day. Control plots were also established for lettuce and chicory, i.e., plots where the conditioning treatment was not applied (MC0 treatment). Brush treatments were performed at the same time each day, in the morning hours. The design of the mechanical conditioning experiment is shown in Table 6 and Figure 3.

### 4.4. Abiotic Parameters of Air and Nutrient Solution

The abiotic parameters of air (minimum and maximum temperature, relative humidity) and nutrient solution (pH, temperature, dissolved oxygen, pH and EC) were measured daily. The average minimum temperature measured in the greenhouses in May was 12.4 °C and the maximum temperature was 31.5 °C, while the relative humidity was 76%. The average pH of the solution was 5.4, while the average EC was 2.2 mS/cm. These conditions during growth resulted in favorable root development. To ensure the supply of sufficient oxygen to the roots, pumps were used which also mixed the nutrient solution. The amount of dissolved oxygen varied considerably during the growing cycle. The highest amount of dissolved oxygen in the nutrient solution (19.3 mg/L) was at the beginning of the growing cycle and the lowest (2.00 mg/L) at the end of the experiment (data not shown).

### 4.5. Harvest Period

Harvesting of lettuce and chicory was done twice during the growing season. The first harvest took place on 28 May, cutting the plants to avoid damaging the vegetative top of the plants and to ensure retro-vegetation. The second harvest took place after 16 days, on 10 June.

### 4.6. Determination of Specialized Metabolites Content

From the group of specialized metabolites following compounds were determined: ascorbic acid (AsA), i.e., vitamin C content, total phenolics (TPC), total flavonoids (TFC) and total non-flavonoids (TNFC). AsA was determined by titration with 2,6-dichlorindophenol according to the standard laboratory method available in AOAC [49]. AsA was isolated from the fresh leaves of lettuce and green chicory with 2% (*v*/*v*) oxalic acid; first 10 g ± 0.01 of fresh plant material was weighed and homogenized with 100 mL of 2% (*v*/*v*) oxalic acid. Prepared solution was filtered through Whatman filter paper and 10 mL of solution was used for titration with 2,6-dichlorindophenol till the appearance of a characteristic pink coloration. Namely, this method is based on the oxidimetric titrations using 2,6-dichlorindophenol as reducing agent. 2,6-dichlorindophenol is a solution of intense blue color that oxidizes L-ascorbic acid to dehydroascorbic acid until the color of the reagent changes, and also serves as an indicator for this redox reaction. The final AsA content was calculated according to Equation (1) and expressed as mg/100 g fresh weight.
AsA = (*V* (DKF) × *F*)/*D* × 100,(1)
where *V* (DKF)—volume of DKF (mL); *F*—factor of DKF; *D*—sample mass used for titration.

The TPC, TFC and TNFC content was determined based on the colorimetric reaction, the development of blue color within phenols and reagent Folin–Ciocalteu measured spectrophotometrically (Shimadzu, 1900i, Kyoto, Japan) at 750 nm using dH_2_O as a blank. The method was described by Ough and Amerine [50]. For the purpose of extraction of polyphenolic compounds from lettuce and green chicory leaves, 10 g ± 0.01 of fresh plant leaves were weighed into an Erlenmeyer flask, and 40 mL of 80% EtOH (*v*/*v*) was added and refluxed. The prepared sample was first heated to boiling point and additionally refluxed for 10 min. After 10 min, the sample was filtered through Whatman filter paper into a volumetric flask of 100 mL. After filtration, the remainder of the sample was transferred to the Erlenmeyer flask, another 50 mL of 80% EtOH (*v*/*v*) was added, and reflux was repeated for another 10 min. After the second reflux, the sample was filtered and the filtrates were combined while the flask was made up to the mark with 80% EtOH (*v*/*v*). The thus prepared sample was subjected to reaction with reagent Folin–Ciocalteu, according to the following procedure: to a volumetric flask of 50 mL, 0.5 mL of the ethanolic plant extract, 30 mL of distilled water (dH_2_O), 2.5 mL of the freshly prepared reagent Folin–Ciocalteu (1:2 with dH_2_O) and 7.5 mL of saturated sodium carbonate solution (Na_2_CO_3_) were added. The flask was made up to the mark with dH_2_O and the reaction was allowed to stand at room temperature for 2 h with intermittent shaking. The same ethanolic extracts prepared for TPC were used for TFC determination. TFC separation was performed according to the following procedure: 10 mL of the ethanolic extract was added to the 25 mL volumetric flask, 5 mL HCl (1:4, *v*/*v*) and 5 mL formaldehyde were added. The prepared samples were treated with nitrogen (N_2_) and left at room temperature for 24 h in a dark place. After 24 h, the samples were filtered and the same reaction was performed with Folin–Ciocalteu reagent as for TPC. Gallic acid was used as external standard and the final concentration of TPC, TFC and TNFC content was expressed as mg GAE/100 g fresh weight. TNFC content was mathematically expressed as the difference between total phenols and flavonoids.

From the plant pigments, the following have been identified: chlorophyll a (Chl_a), chlorophyll b (Chl_b), total chlorophylls (TCh) and total carotenoids (TCA), according to the method described by Holm [51] and Wettstein [52]. For the extraction of pigments from leaves of lettuce and green chicory, 0.2 g ± 0.01 of fresh plant leaves were weighed and a total volume of 15 mL of acetone (p.a.) was added, a total of three times. After each addition of acetone, the samples were homogenized using a laboratory homogenizer (IKA, UltraTurax T-18, Staufencity, Germany). The final solution was filtered and transferred to a 25 mL volumetric flask. The absorbance was measured spectrophotometrically (Shimadzu UV 1900i, Kyoto, Japan) at three wavelengths, 662, 644 and 440 nm, using acetone as a blank. The equations of Holm–Wettstein were used to quantify the individual pigments (2), and the final content was expressed in mg/g.
Chl_a = 9.784 × *A*_662_ − 0.990 × *A*_644_ [mg/L]Chl_b = 21.426 × *A*_644_ − 4.65 × *A*_622_ [mg/L]TCh = 5.134 × *A*_662_ + 20.436 × *A*_644_ [mg/L]TCA = 4.695 × *A*_440_ − 0.268 × TCh [mg/L](2)

### 4.7. Determination of Antioxidant Capacity

For the determination of antioxidant capacity, the ABTS assay was performed [53]; ABTS, 2,2′-azinobis (3-ethylbenzothiazoline-6-sulfonic acid), potassium persulfate, and Trolox were obtained from Sigma-Aldrich (St. Louis, MO, USA). Trolox (6-hydroxy-2,5,7,8-tetramethylchroman-2-carboxylic acid) was used as the antioxidant standard, and a stock standard Trolox (2.5 mM) was prepared in ethanol (80% *v*/*v*). To prepare the ABTS radical solution (ABTS+), 5 mL of ABTS solution (7 mM) and 88 mL of potassium persulfate solution (140 mM) were mixed and allowed to stand for 16 h in the dark at room temperature. On the day of analysis, a 1% ABTS+ solution (in 96% ethanol) was prepared. A total of 160 µL of ethanolic extract (prepared for phenol isolation) was directly injected into the cuvette and mixed with 2 mL of 1% ABTS+ while absorbance was measured at 734 nm (Shimadzu 1900i, Kyoto, Japan). The final results of antioxidant capacity were calculated from the calibration curve and expressed as µmol TE/L.

### 4.8. Statistical Analysis

Each sample (cultivar, mechanical conditioning treatment and control) of leafy vegetables cultivated in floating hydroponics was represented by two boards, while all chemical laboratory analyzes were performed in triplicate. The data obtained were averaged, expressed as mean ± standard deviation (SD), as shown in the figures and tables. The ANOVA and Duncan’s multiple range tests (95% confidence limit) were performed to show the variations in the mean values among the samples. SAS statistical software ver. 9.4. was used for this purpose [54]. Different letters show significant differences between the means at *p* ≤ 0.0001, while also the average deviation of the results from the mean for each parameter studied is expressed with the values of standard deviation.

## 5. Conclusions

The mechanically-induced stress in the form of brushing per day (10 and 20) did not cause damage to plant tissue and thus did not significantly affect the processes of primary metabolism, i.e., photosynthesis, as shown by the higher contents of total chlorophylls and carotenoids in lettuce in both harvest periods; with an average of 22% higher TCh in plants treated with 20 brushings per day in the first harvest period and 18% higher TCh in the second harvest period compared to the non-treated plants, and 33% higher TCA in the first and 24% higher in the second harvest period also in plants treated with 20 brushings per day compared to the non-treated plants. Moreover, the induced mechanical stimuli were sufficient to initiate plant signaling molecules for stress defense, as evidenced by higher levels of antioxidants such as ascorbic acid and polyphenolic compounds in the plants treated with mechanical conditioning. For lettuce in the first harvest period, AsA content was on average 55% higher in mechanically-stimulated plants compared to the non-treated plants, while for green chicory, a more pronounced effect of MIS was observed in the second harvest period, in which, on average, 24% higher AsA content was determined in treated plants. Polyphenolic compounds in the first harvest period in lettuce were on average 88% higher in MIS treated plants, regardless of the passes per day, while in the second harvest period, about 11% higher polyphenolic compounds were determined in comparison with non-treated plants. From all these, it can be concluded that implementation of induced mechanical stress is a good practice in the cultivation of leafy vegetables, the application of which produces high quality plant material with high nutritional potential, and significantly higher levels of antioxidants and phytochemicals important for human health. It should also be emphasized that most authors explain the effects of some abiotic stresses, mostly drought, high temperatures, salinity, etc., but there is still a lack of scientific data on the effects of mechanically-induced stress on the phytochemical status of plants. Individual compound determination is necessary to further explain the effects of mechanically-induced stress on the plant organism.

## Figures and Tables

**Figure 1 plants-10-02650-f001:**
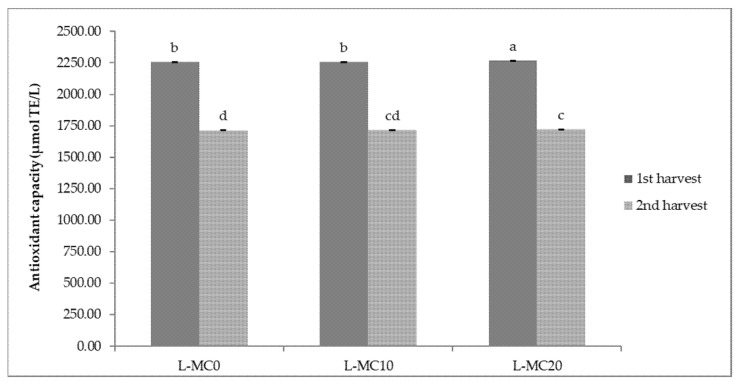
Antioxidant capacity of lettuce depending on different mechanical conditioning treatments. L—lettuce; MC—mechanical conditioning; 0, 10, 20—number of plants brushing per day. Different letters show significant statistical differences between means.

**Figure 2 plants-10-02650-f002:**
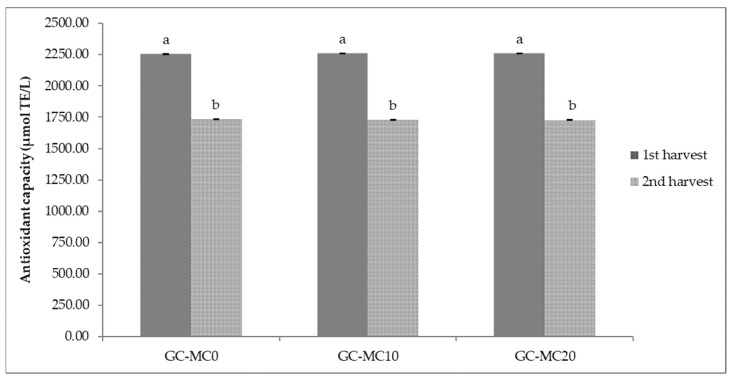
Antioxidant capacity of green chicory depending on different mechanical conditioning treatments. GC—green chicory; MC—mechanical conditioning; 0, 10, 20—number of plants brushing per day. Different letters show significant statistical differences between means.

**Figure 3 plants-10-02650-f003:**
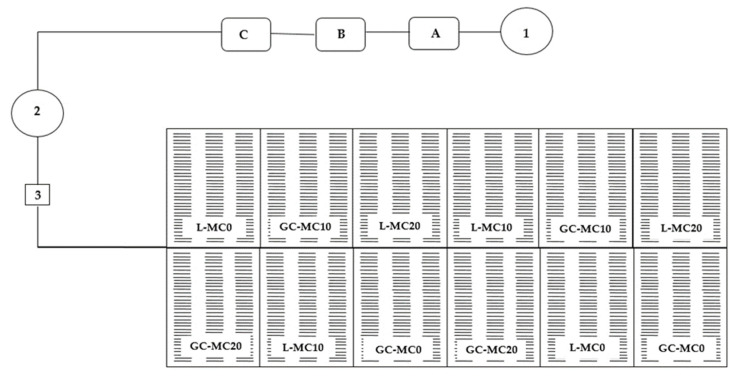
Graphical scheme of floating hydroponics cultivation of lettuce and green chicory. 1—water; A, B, C—tanks for concentrated nutrient solutions and injectors; 2—tanks for standard nutrient solutions; 3—pump; L—lettuce; GC—green chicory; MC—mechanical conditioning; 0, 10, 20—number of plants brushing per day.

**Table 1 plants-10-02650-t001:** Content of specialized metabolites in lettuce under the influence of brushing.

Treatment	AsA (mg/100 g fw)	TPC (mg GAE/100 g fw)	TFC (mg GAE/100 g fw)	TNFC (mg GAE/100 g fw)
1st harvest				
L-MC0_1_	15.53 ± 1.20 ^b^	51.38 ± 5.67 ^e^	35.59 ± 1.03 ^f^	15.77 ± 6.50 ^d^
L-MC10_1_	23.71 ± 3.20 ^a^	90.24 ± 0.82 ^d^	55.19 ± 0.47 ^d^	35.06 ± 1.27 ^c^
L-MC20_1_	24.39 ± 2.03 ^a^	102.46 ± 1.25 ^c^	66.72 ± 0.43 ^b^	35.74 ± 1.14 ^c^
2nd harvest				
L-MC0_2_	16.45 ± 2.35 ^b^	107.47 ± 0.43 ^b^	44.30 ± 0.29 ^e^	90.34 ± 0.51 ^a^
L-MC10_2_	17.60 ± 2.52 ^b^	150.94 ± 1.52 ^a^	60.60± 0.1 ^c^	63.17 ± 1.36 ^b^
L-MC20_2_	15.90 ± 1.97 ^b^	86.89 ± 0.57 ^d^	70.75 ± 0.67 ^a^	16.13 ± 1.14 ^d^
ANOVA	*p* ≤ 0.0008	*p* ≤ 0.0001	*p* ≤ 0.0001	*p* ≤ 0.0001

AsA—ascorbic acid content; TPC—total phenolics content; TNFC—total non-flavonoid content; TFC—total flavonoid content. Different letters show significant statistical differences between means.

**Table 2 plants-10-02650-t002:** Pigment compounds of lettuce under the influence of brushing.

Treatment	Chl_a (mg/g)	Chl_b (mg/g)	TCh (mg/g)	TCA (mg/g)
1st harvest				
L-MC0_1_	0.2 ± 0.004 ^d^	0.16 ± 0.002 ^a^	0.36 ± 0.006 ^e^	0.15 ± 0.002 ^e^
L-MC10_1_	0.17 ± 0.001 ^e^	0.12 ± 0.002 ^b^	0.29 ± 0.002 ^f^	0.1 ± 0.001 ^f^
L-MC20_1_	0.28 ± 0.001 ^c^	0.15 ± 0.001 ^ab^	0.44 ± 0.001 ^d^	0.20 ± 0.001 ^d^
2nd harvest				
L-MC0_2_	0.28 ± 0.002 ^c^	0.17 ± 0.001 ^a^	0.45 ± 0.002 ^c^	0.21 ± 0.001 ^c^
L-MC10_2_	0.37 ± 0.002 ^a^	0.17 ± 0.045 ^a^	0.57 ± 0.003 ^a^	0.27 ± 0.001 ^a^
L-MC20_2_	0.35 ± 0.001 ^b^	0.18 ± 0.001 ^a^	0.53 ± 0.002 ^b^	0.26 ± 0.001 ^b^
ANOVA	*p* ≤ 0.0001	*p* ≤ 0.0240	*p* ≤ 0.0001	*p* ≤ 0.0001

Chl_a—chorophyll a content; Chl_b—chorophyll b content; TCh—total chlorophyll content; TCA—total carotenoids. Different letters show significant statistical differences between means.

**Table 3 plants-10-02650-t003:** Content of specialized metabolites in green chicory under the influence of brushing.

Treatment	AsA (mg/100 g fw)	TPC (mg GAE/100 g fw)	TFC (mg GAE/100 g fw)	TNFC (mg GAE/100 g fw)
1st harvest				
GC-MC0_1_	36.52 ± 2.75 ^b^	104.85 ± 1.60 ^d^	42.33 ± 1.95 ^d^	62.51 ± 0.50 ^c^
GC-MC10_1_	40.97 ± 0.77 ^a^	79.42 ± 1.13 ^e^	26.80 ± 2.20 ^e^	52.61 ± 1.27 ^d^
GC-MC20_1_	34.45 ± 4.47 ^b^	50.03 ± 1.48 ^f^	16.92 ± 1.84 ^f^	33.09 ± 0.35 ^e^
2nd harvest				
GC-MC0_2_	23.64 ± 1.97 ^c^	172.64 ± 0.28 ^b^	96.35 ± 0.72^b^	76.28 ± 0.50 ^a^
GC-MC10_2_	25.31 ± 0.74 ^c^	161.06 ± 0.69 ^c^	92.45 ± 1.08^c^	68.60 ± 0.39 ^b^
GC-MC20_2_	33.23 ± 0.75 ^b^	179.77 ± 0.62 ^a^	103.14 ± 1.33^a^	76.63 ± 0.82 ^a^
ANOVA	*p* ≤ 0.0001	*p* ≤ 0.0001	*p* ≤ 0.0001	*p* ≤ 0.0001

AsA—ascorbic acid content; TPC—total phenolics content; TNFC—total non-flavonoid content; TFC—total flavonoid content. Different letters show significant statistical differences between means.

**Table 4 plants-10-02650-t004:** Pigment compounds of green chicory under the influence of brushing.

Treatment	Chl_a (mg/g)	Chl_b (mg/g)	TCh (mg/g)	TCA (mg/g)
1st harvest				
GC-MC0_1_	0.36 ± 0.001 ^e^	0.15 ± 0.001 ^f^	0.51 ± 0.001 ^b^	0.25 ± 0.001 ^e^
GC-MC10_1_	0.43 ± 0.001 ^d^	0.20 ± 0.001 ^d^	0.63 ± 0.002 ^b^	0.30 ± 0.001 ^d^
GC-MC20_1_	0.33 ± 0.001 ^f^	0.18 ± 0.001 ^e^	0.51 ± 0.001 ^b^	0.23 ± 0.001 ^f^
2nd harvest				
GC-MC0_2_	0.61 ± 0.002 ^b^	0.29 ± 0.002 ^c^	0.89 ± 0.004 ^a^	0.42 ± 0.002 ^b^
GC-MC10_2_	0.71 ± 0.001 ^a^	0.36 ^a^	0.92 ± 0.26 ^a^	0.48 ± 0.02 ^a^
GC-MC20_2_	0.56 ^c^	0.30 ^b^	0.86 ± 0.001 ^a^	0.39 ± 0.001 ^c^
ANOVA	*p* ≤ 0.0001	*p* ≤ 0.0001	*p* ≤ 0.0008	*p* ≤ 0.0001

Chl_a—chorophyll a content; Chl_b—chorophyll b content; TCh—total chlorophyll content; TCA—total carotenoids. Different letters show significant statistical differences between means.

**Table 5 plants-10-02650-t005:** Adapted nutrient solution for lettuce and green chicory cultivation.

Biogenic Element	Measuring Unit	Values
pH	mS/cm	5.5
EC		2.5
Na	mmol/L	<6
Cl		<6
HCO_3_		<0.5
N-NH_4_	mmol/L	<0.5
K		6
Ca		6
Mg		2
N-NO_3_	mmol/L	14
S		2
P		2
Fe	μmol/L	40
Mn		8
Zn		8
B		50
Cu		1.5
Mo		1.5

**Table 6 plants-10-02650-t006:** The design of the mechanical conditioning experiment.

Species	Brushing	Hydroponic System	Harvest Period	Treatment
Lettuce	0	FH	I.	L-MC0_1_
			II.	L-MC10_2_
	10	FH	I.	L-MC20_1_
			II.	L-MC0_2_
	20	FH	I.	L-MC10_1_
			II.	L-MC20_2_
Green chicory	0	FH	I.	GC-MC0_1_
			II.	GC-MC10_2_
	10	FH	I.	GC-MC20_1_
			II.	GC-MC0_2_
	20	FH	I.	GC-MC10_1_
			II.	GC-MC20_2_

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
