# Peer review of "Plant Response to Mechanically-Induced Stress: A Case Study on Specialized Metabolites of Leafy Vegetables"

_plants, 2021, doi:10.3390/plants10122650_

Round 1

Reviewer 1 Report

This is an original research article, which highlights the influence of mechanical stimulation (brushing) on the ascorbic acid, total phenolics, total non-flavonoid, total flavonoid, chlorophyll content, and antioxidant capacity of lettuce and green chicory plants.

The topic presents interest because it analyses lettuce and green chicory, some very used leafy vegetables. 

Some suggestions for the authors:

  1. The introduction part is well written.
  2. Please include in the manuscript the year in which the study was conducted.
  3. In the material and method section part, it is not very clear on which plants the experiment was conducted ('Reggina di Maggio' 25%; 'Meraviglia delle Quatro Stagioni'25%; 'Grunetta' 25%; 'Cavolo di Napoli' 25% (Hortus sementi, Italy), line 293, 294. A mixture of seeds was sown, and the parameters were measured on which plant? If it was presented as a mean (eg arithmetic), please mention it in the article text.
  4. The Results part is well written, and the figures/tables are appropriate and easy to understand.
  5. The Discussion part needs improvement. Please include in this part your findings and correlate them to other published results.
  6. The latest articles published on lettuce and green chicory plants should be introduced and correlated with the results obtained in this study.
  7. The Conclusion part needs to be rewritten, to summarise the obtained results.

Author Response

Manuscript ID: plants-1476930

Manuscript title:  Plant response to mechanically induced stress - a case study on specialized metabolites of leafy vegetables 

Dear respectable Reviewers,

Thank you for your valuable suggestions. We have reviewed the spelling and also checked the entire text for English language and style and made some corrections.

We have also used the "Track Changes" option and by yellow colour we highlighted the parts of the text we have corrected according to your valuable suggestions.

Reviewer 1 comments:

  1. Please include in the manuscript the year in which the study was conducted.

RESPONSE: Line 289- Authors added a year in which study was conducted.

  1. In the material and method section part, it is not very clear on which plants the experiment was conducted ('Reggina di Maggio' 25%; 'Meraviglia delle Quatro Stagioni'25%; 'Grunetta' 25%; 'Cavolo di Napoli' 25% (Hortus sementi, Italy), line 293, 294. A mixture of seeds was sown, and the parameters were measured on which plant? If it was presented as a mean (eg arithmetic), please mention it in the article text.

RESPONSE: The parameters were measured on the species lettuce (Lactuca sativa L.). This is a commercially available mixed lettuce seed packet containing the named cultivars in the proportions indicated.

  1. The Discussion part needs improvement. Please include in this part your findings and correlate them to other published results.

RESPONSE: Considering that the research related to mechanically induced stress is specific and there is a general lack of scientific research on specific vegetable species such as lettuce and green chicory, the authors, in accordance with the Reviewer suggestion, have improved the discussion by presenting recent results and data on nutrient composition, i.e., the composition of specific lettuce and chicory metabolites (Line 208-216; Line 251-257; Line 287-291; Line 305-307).

  1. The latest articles published on lettuce and green chicory plants should be introduced and correlated with the results obtained in this study.

RESPONSE: in the parts of the discussion as suggested we correlate the latest published articles and results obtained within this research.

  1. The Conclusion part needs to be rewritten, to summarise the obtained results.

RESPONSE: we rewritten the conclusion.

Reviewer 2 Report

The research conducted by the authors is interesting, the results are presented properly and the discussion is good enough. Generally, this manuscript is well written. In the Introduction part the authors explained that still is a little known about mechanically induced stress on the content of specialized metabolites, or phytochemicals with significant antioxidant activity of leafy vegetables, especially lettuce and green chicory. Accordingly, the authors underlined the aim of this study, determination of specialized metabolites content and antioxidant capacity of lettuce and green chicory under the influence of mechanical stimulation by brushing. The results of this work are nicely described and discussed properly with available literature data. In Material and methods part the authors prepared very detailed protocols for every single step of experimental work in this article. Statistical analyses were also described in detail.

The main remark is related right to statistical analyses. On Figure 1 the authors presented the results considering antioxidant capacity of lettuce and green chicory. Firstly, it is not clear if the results of lettuce and green chicory were compared together for both plant species in both harvest period or separately for each plant species. If so, and the results were compared together for both plant species why all other parameters (AsA, TPC, TFC, TNFC and pigment content) were not compare together? Also, in Line 155 the authors stated that high antioxidant capacity was observed in green chicory in both, the first and the second harvest period but it is not clear in comparison to what? On the other hand, the authors in the line 145 described that antioxidant capacity of both plant species, in general, were higher in the first harvest period than in the second. Obviously it is true, but why the same letter “b” showing no significant difference between means is placed on the samples collected from the first harvest (L-MC0 and L-MC 10) and from the second harvest (GC-MC0, GC-MC10 and GC-MC20)? It is confusing and hard to understand. Accordingly, it is not clear enough if the goal of this experimental work was to compare lettuce and green chicory under mechanical stress or to investigate the effect of this stress on these two species separately. In Tables 1-4 the authors presented the results separately but in the Figure 1 it seems that the results are presented for both plant species together but these need to be unique.

I also have several minor remarks which I assigned directly in the text manuscript.

Author Response

Manuscript ID: plants-1476930

Manuscript title:  Plant response to mechanically induced stress - a case study on specialized metabolites of leafy vegetables 

Dear respectable Reviewers,

Thank you for your valuable suggestions. We have reviewed the spelling and also checked the entire text for English language and style and made some corrections.

We have also used the "Track Changes" option and by yellow colour we highlighted the parts of the text we have corrected according to your valuable suggestions.

Reviewer 2 comments:

GENERAL COMMENT:

The main remark is related right to statistical analyses. On Figure 1 the authors presented the results considering antioxidant capacity of lettuce and green chicory. Firstly, it is not clear if the results of lettuce and green chicory were compared together for both plant species in both harvest period or separately for each plant species. If so, and the results were compared together for both plant species why all other parameters (AsA, TPC, TFC, TNFC and pigment content) were not compare together? Also, in Line 155 the authors stated that high antioxidant capacity was observed in green chicory in both, the first and the second harvest period but it is not clear in comparison to what? On the other hand, the authors in the line 145 described that antioxidant capacity of both plant species, in general, were higher in the first harvest period than in the second. Obviously, it is true, but why the same letter “b” showing no significant difference between means is placed on the samples collected from the first harvest (L-MC0 and L-MC 10) and from the second harvest (GC-MC0, GC-MC10 and GC-MC20)? It is confusing and hard to understand. Accordingly, it is not clear enough if the goal of this experimental work was to compare lettuce and green chicory under mechanical stress or to investigate the effect of this stress on these two species separately. In Tables 1-4 the authors presented the results separately but in the Figure 1 it seems that the results are presented for both plant species together but these need to be unique.

RESPONSE: The species studied, lettuce and chicory, were not statistically compared, but the main variables, i.e., factors in the experiment were mechanical conditioning (brushing passes 0, 10 and 20 times per day) and harvest period, and in this context the authors agree that Figure 1 does not adequately represent the results on antioxidant capacity. The authors have changed the version of Figure 1 according to the suggestions and presented the antioxidant capacity results for the lettuce and green chicory samples in two separate figures (Figure 1 and 2). In addition, the authors carefully reviewed the statistical analysis and corrected the letters indicating significant differences in Figure 1.

SPECIFIC COMMENTS:

RESPONSE: Authors corrected and indicated the corrected parts of the text in the main text.

Reviewer 3 Report

The manuscript " Plant response to mechanically induced stress - a case study on specialized metabolites of leafy vegetables" deals with a study that could be of interest for a broader audience, however the quality of the manuscript is below standard.

The authors did not respect some basics of scientific publishing, when something is not significant different, then it is not different. On the other side when a general statement is true for only a part of the data, then the statement should not be made. Furthermore, in scientific writing most things have a clear definition, a word/term has a known, generally accepted meaning, this must be respected otherwise confusion is created. Thirdly, when making a difference between two terms with a different unit, the result is a number without any meaning, the difference between three apples and two pears is not one apple nor minus one pear, it is just nothing meaningful. 

Finally the discussion is far too long, there is a very limited set of data and by turning the data round and around the main message that could be in the data is lost in comparisons that make no sense since not supported by data. Furthermore by producing more and more text the authors accumulate more and more errors. A short discussion of 500 words would be sufficient in this case. I did not read the text till the end, sorry. 

I hope that these comments and the remarks added in the attached document can be a guide for the authors to improve the manuscript, which may become a nice small paper in a journal focused on agriculture. However for the moment there can only be one recommendation and that is to reject the manuscript for publication.   

Author Response

Manuscript ID: plants-1476930

Manuscript title:  Plant response to mechanically induced stress - a case study on specialized metabolites of leafy vegetables 

Dear respectable Reviewers,

Thank you for your valuable suggestions. We have reviewed the spelling and also checked the entire text for English language and style and made some corrections.

We have also used the "Track Changes" option and by yellow colour we highlighted the parts of the text we have corrected according to your valuable suggestions.

Reviewer 3 comments:

GENERAL COMMENTS:

The authors did not respect some basics of scientific publishing, when something is not significant different, then it is not different. On the other side when a general statement is true for only a part of the data, then the statement should not be made. Furthermore, in scientific writing most things have a clear definition, a word/term has a known, generally accepted meaning, this must be respected otherwise confusion is created. Thirdly, when making a difference between two terms with a different unit, the result is a number without any meaning, the difference between three apples and two pears is not one apple nor minus one pear, it is just nothing meaningful.

Finally the discussion is far too long, there is a very limited set of data and by turning the data round and around the main message that could be in the data is lost in comparisons that make no sense since not supported by data. Furthermore by producing more and more text the authors accumulate more and more errors. A short discussion of 500 words would be sufficient in this case. I did not read the text till the end, sorry.

RESPONSE: We thank you for your valuable opinion. Authors based on the results obtained in the experiment emphasize and discuss issues that consider the effects of mechanically induced stress from as many angles as possible. We agree that the discussion in some sections is extensive, but again, the authors have expanded the discussion in this sense to provide a sound basis for understanding stress, i.e., all the mechanisms by which the plant responds to stress.

The author team is extremely sorry that unfortunately you were not able to read all parts and aspects of the discussion because ultimately it connects all segments that were important to us to give scientific readers as many recent results related to better understanding of the effects of mechanical stress and ultimately new practices in the production of leafy vegetable species, which allows us to produce a nutritiously higher quality product.

SPECIFIC COMMENTS:

RESPONSE: The authors have carefully considered all of your comments in the text of the paper and have rewritten according to the suggestions where they considered applicable.

Round 2

Reviewer 1 Report

The article is improved and can be published in its present form.

Only minor editing of the English language and style is required.

Author Response

Dear respectable Reviewer,

Thank you for your valuable suggestions. We have reviewed the spelling and also checked the entire text for English language and style and made some corrections.

We have also used the "Track Changes" option and by yellow colour we highlighted the parts of the text we have corrected according to your valuable suggestions.

Reviewer 2 Report

The revised version of submitted paper addressed to all reviewers concerns in details and this revised and improved manuscript according to reviewer suggestions is now acceptable for publication.

Author Response

Dear respectable Reviewer,

Thank you for your valuable review.

Best Regards, Sanja Radman

Reviewer 3 Report

The authors took less than two days to make the changes that would turn a "reject" into an "accepted, when doing things seriously this is not possible.

I invite the authors to read some sentences in the material and method section which they changed so that the issue of adding GAE with CTH equivalents was solved. First of all, please explain to me now what you did exactly, because this is now more unclear than before. Here are the sentences "The absorbance of blue color in both TPC and TFC reactions was measured spectro-photometrically (Shimadzu, 1900i, Germany) at 750 nm using dH2O as a blank. Gallic acid and catechol werewas used as external standards and the concentration of TPC, and TFC and TNFC content was expressed as mg GAE/100 g fresh weight. and mg CTH/100 mg fresh weight, respectively. TNFC content was mathematically expressed as the difference between total phenols and flavonoids." so by changing the names the problem is solved? I do not agree. Either the authors explain what they did exactly and say that they did not use catechol as reference or the column with TNFC needs to be eliminated from the manuscript. 

Why do I consider the discussion too long. Take for instance the lines 305-310, the auhors use a study as compsrison to say MIS results in an increase in chlorophyll but this is not the case actually. L-MC10-1 has a lower TCh than the LMC0-1, and for L-MC-2 there is a peak for 10 strokes.

Now looking at the GC samples, for GC-MC-1 there is again a peak with 10 strokes but after 20 strokes the TCh is at the same level as the control. Now for GC-MC-2 there is an increase with 10 strokes but after 20 strokes the THc is lower than control. Of course keeping in mind that stroking has no impact on THc since none of these changes are significant for GC samples.  

So basically there is nothing to discuss for the GC samples, stroking has no impact on TCh, for the lettuce samples the image is much more complex than "TCh was highest in the samples treated with mechanical conditioning".

My recommendation is therefore the same as before

Author Response

Manuscript ID: plants-1476930

Manuscript title:  Plant response to mechanically induced stress - a case study on specialized metabolites of leafy vegetables 

Dear respectable Reviewer,

Thank you for your suggestions.

  1. Materials and methods.

RESPONSE: We used a gallic acid as an external standard for both total phenol and flavonoid reaction. We overlooked the error in writing abbreviations and thank you for noticing.

  1. Results and discussion.

RESPONSE: We have marked some changes as suggested. Also, in the Results section we have clearly stated all the results obtained, while in the Discussion section we have focused more on how to explain certain phenomena in the plant organism as a function of MIS.

Thank you for your valuable time!

Best regards, in front of the Authors' team

                                                                                                 Sanja Radman

November 26, 2021
